# Ebselen as template for stabilization of A4V mutant dimer for motor neuron disease therapy

Varunya Chantadul[1,2], Gareth S.A. Wright [1], Kangsa Amporndanai[1], Munazza Shahid[3], Svetlana V. Antonyuk [1], Gina Washbourn[4], Michael Rogers[4], Natalie Roberts[4], Matthew Pye[4], Paul M. O'Neill[4] & S. Samar Hasnain [1✉]

Mutations to the gene encoding superoxide dismutase-1 (SOD1) were the first genetic elements discovered that cause motor neuron disease (MND). These mutations result in compromised SOD1 dimer stability, with one of the severest and most common mutations Ala4Val (A4V) displaying a propensity to monomerise and aggregate leading to neuronal death. We show that the clinically used ebselen and related analogues promote thermal stability of A4V SOD1 when binding to Cys111 only. We have developed a A4V SOD1 differential scanning fluorescence-based assay on a C6S mutation background that is effective in assessing suitability of compounds. Crystallographic data show that the selenium atom of these compounds binds covalently to A4V SOD1 at Cys111 at the dimer interface, resulting in stabilisation. This together with chemical amenability for hit expansion of ebselen and its on-target SOD1 pharmacological chaperone activity holds remarkable promise for structure-based therapeutics for MND using ebselen as a template.

[1] Faculty of Health and Life Sciences, Molecular Biophysics Group, Institute of Integrative Biology, University of Liverpool, Liverpool L69 7ZB, UK. [2] Faculty of Dentistry, Department of Anatomy, Mahidol University, Bangkok 10400, Thailand. [3] Department of Chemistry and Chemical Engineering, Lahore University of Management Sciences, Punjab 54792, Pakistan. [4] Faculty of Science and Engineering, Department of Chemistry, University of Liverpool, Liverpool L69 7ZD, UK. ✉email: S.S.Hasnain@liverpool.ac.uk

Amyotrophic lateral sclerosis (ALS), also referred to as motor neuron disease (MND) and Lou Gehrig's disease, is characterised by progressive degeneration of motor neurons in the cerebral cortex and spinal cord. ALS patients typically present with focal muscular weakness and progress to paralysis of the diaphragm, which leads to respiratory failure and death usually within 5 years of the onset of symptoms[1]. Although approximately 90% of ALS cases are sporadic (sALS), the remaining 10% of cases arise from inheritance of mutated genes (familial type ALS; fALS), which are generally the cause of early onset ALS[1,2]. The gene encoding Cu, Zn superoxide dismutase (SOD1) was the first gene to be discovered as an ALS-linked genetic factor in 1993[3]. An alanine-to-valine mutation at amino acid position 4 within the SOD1 protein (A4V), accounting for 50% of SOD1-related gene mutations in the US, causes a rapid and aggressive form of fALS, thereby reducing survival time to less than 2 years[4].

SOD1 is a homodimeric copper-containing and zinc-containing enzyme, serving as one of the cellular protective mechanisms against reactive oxygen species[5]. However, SOD1-fALS does not result from a malfunction in its enzymatic dismutase activity. Instead, an increase in the aggregation propensity of mutant SOD1 is thought to lead to disruption of physiological homoeostasis, and has been considered a gain of function that leads to fALS[6]. Previous studies suggested that SOD1 dimer destabilisation and dissociation are critical for aggregation[7–10]. As proof of principle, insertion of an inter-subunit disulphide linkage by genetic modification inhibited SOD1 aggregation[8], and several small molecules have been shown to replicate this effect[11–13]. With regard to its location at the solvent-accessible dimer interface and its nucleophilicity, Cys111 has been a particular focus of drug binding studies. This residue was demonstrated to be the site of maleimide-derived compound binding in G93A and G85R SOD1 which promoted thermostability[11]. Binding of cisplatin to Cys111 also increases thermal stability, inhibits oligomerisation and solubilises oligomers of wild-type metal free (apo) SOD1[12]. However, both compounds hinder the maturation of SOD1 by preventing complex formation between SOD1 and its copper chaperone (hCCS)[14].

Ebselen is an organoselenium compound with activity similar to glutathione peroxidase[15]. Several lines of evidence have demonstrated the neuroprotective effects of ebselen, possibly via its antioxidant properties[16–18]. The ability of ebselen to reduce mitochondrial cellular toxicity generated by mutant SOD1 indicates this compound could play an important role in mitigating fALS[19]. We recently demonstrated crystallographically that ebselen binds SOD1 at Cys111 and leads to an increase in dimer affinity of wild-type and several SOD1 mutants without disruption of SOD1-hCCS complex[13]. We also showed that ebselen promotes formation of the Cys57-Cys146 disulphide bond of G93A and A4V SOD1 in human cells[13]. Here we introduce an innovative derivative series of ebselen and ebsulphur, a sulphur analogue of ebselen, that have been designed to improve pharmacological effects. To evaluate the performance of potential lead compounds, a differential scanning fluorescence (DSF) assay using C6S background SOD1 has been developed creating an effective medium-throughput screen reporting on binding at Cys111. Ebselen-based compounds, especially with molecular weights below 400 Dalton, and some ebsulphur derivatives promote thermal stability of A4V SOD1 in this DSF assay. Moreover, we demonstrate that ebselen and a number of its analogues form a covalent bond with Cys111 of A4V SOD1 using x-ray crystallography. The combination of DSF assays on C6S background SOD1 and co-crystallographic structures of compounds with A4V SOD1 provide clear evidence of SOD1 thermal stabilisation and offer opportunities for rational design for the next generation of lead compounds.

## Results

**Synthesis of ebselen and ebsulphur analogues.** In terms of rationale for the analogues prepared, compound **1** was designed to potentially enhance the cysteine reactivity of the ebselen core. Compound **2** was designed to enhance aqueous stability of **1** by incorporation of a basic benzyl morpholine ring system; Compounds **3**, **4** and **5** are simple benzyl analogues which have greater flexibility than the parent drug substituted with electron donating (−OMe) and with electron withdrawing (−Cl) groups. Compounds **6** and **7** are biaryl analogues designed to enhance the potential membrane permeability/penetration with **8**–**10** further modified to incorporate protonatable nitrogens as more aqueous soluble versions of **6** and **7**. Also included in our analysis was a series of ebsulphur (**11**) analogues (**12**, **13**, **16**–**20**) including unreactive amide analogue (**14**) and phthalimide (**15**) as controls to monitor the importance of a cysteine reactive warhead.

Figure 1 contains key data for the series (ClogP, ClogD, MW, TPSA) along with data generated by the Pfizer central nervous system multiparameter optimisation (CNS MPO) algorithm with CNS MPO desirability scores[20]. The MPO scores for compounds in Fig. 1 range from 2.6 to 6.0; compounds with a score of 4.0 or higher are deemed to be more desirable and are predicted to be more likely to reach their CNS drug target compared to compounds with scores of 3.9 or less. The Pfizer CNS MPO tool has helped increase the percentage of compounds nominated for clinical development that exhibit alignment of ADME attributes, cross the blood-brain barrier, and result in lower-risk drug safety (low ClogP and high TPSA chemical space). As can be seen from this data the ebselen analogues **1**–**5** have excellent properties with **11**, **12**, **19** and **20** possessing the best all round properties form the ebsulphur series.

**Compounds that promote thermal stability of A4V SOD1.** There are two free cysteines in mature human SOD1. Cys111 is located at the solvent accessible dimer interface and Cys6 is located in β-strand 1 with side-chain projecting into the β-barrel with limited access to solvent in the folded conformation. However, our DSF study on wild-type and A4V SOD1 showed a biphasic melting curve or decreases in the melting temperature when the proteins were incubated with ebselen overnight and the excess ligands were removed by dialysis (Fig. 2a). The same results were obtained from the wild-type and A4V SOD1 incubated with compounds **1**, **2**, **6** and ebsulphur (Fig. 2a–d, and Table 1). This might be because some of the compounds bind not only at Cys111 but also interact with Cys6 in vitro as we demonstrate in Supplementary Fig. 1 that C111S SOD1 with ebselen shows the biphasic melting curve with the first melting temperature ($T_m$) decreased 10 °C from the control. We thus created a C6S mutant for the wild type and A4V mutant removing the possibility of some proportion of the compound binding to Cys6. DSF studies were performed on the C6S background for both wild-type and A4V SOD1 (WT[C6S] and A4V[C6S]). The shifts in $T_m$ of the proteins conjugated with ebselen-based and ebsulphur-based derivatives are presented in Table 1. The results show that most of the ebselen-based compounds increase the $T_m$ of WT[C6S] and A4V[C6S] notably except compounds **6** and **7**, which decreased the $T_m$ (Fig. 2e, f). A4V[C6S] bound to the lower molecular mass compounds (ebselen, and compounds **1**–**5**) exhibits 4–8 °C increase in $T_m$, while the protein bound to the larger hydrophobic compounds (compounds **8**, **9** and **10**) shows a modest increase between 1–4 °C in $T_m$. The same trend was also observed in WT[C6S], albeit less pronounced as this pseudo-WT protein is highly stable. We note that the increase in $T_m$ by ebselen, compounds **1** and **2** could not be observed in the wild-type and A4V SOD1 without C6S mutation.

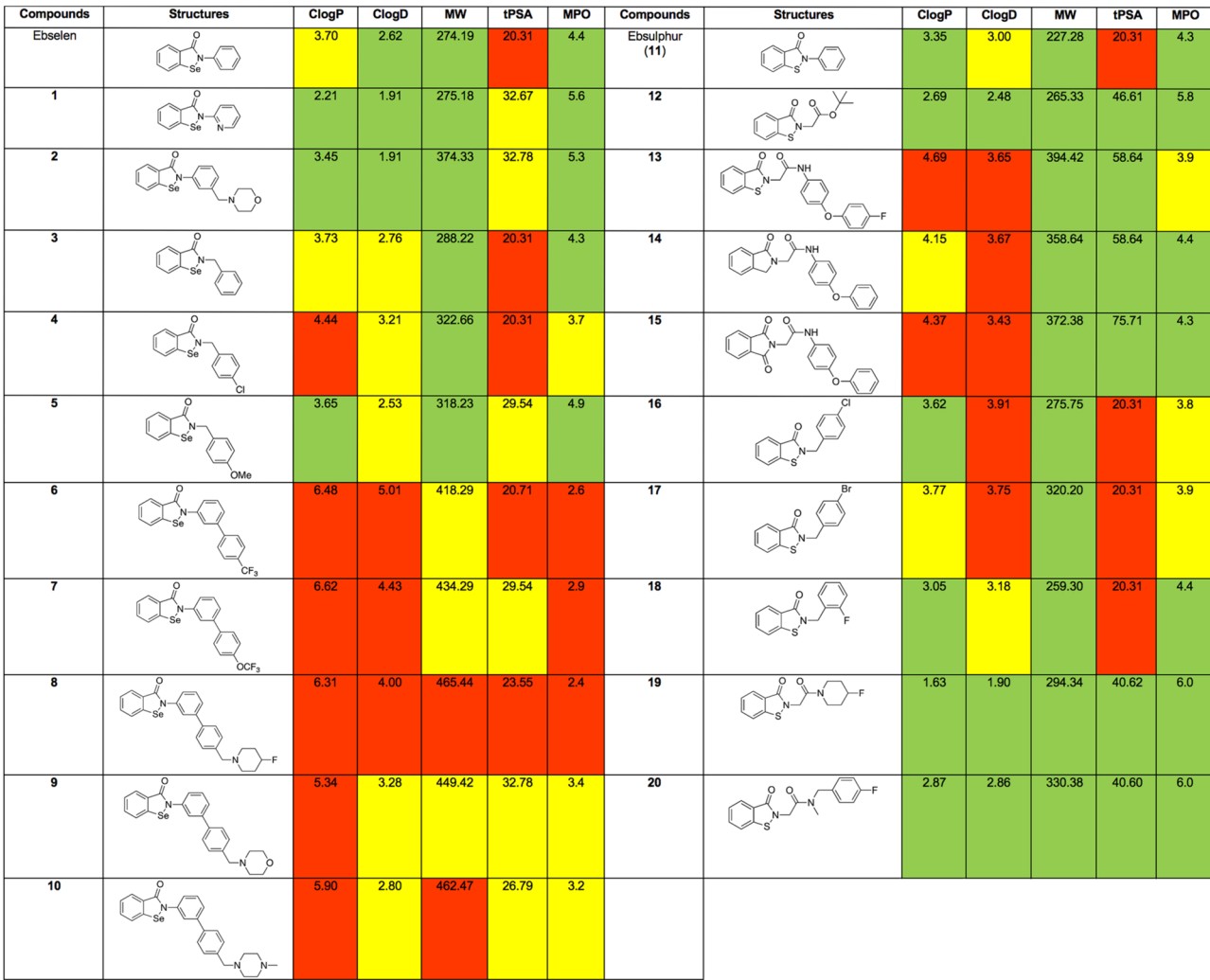

**Fig. 1 Calculated physicochemical properties of series including CNS multiparameter optimisation score.** The values are colour coded according to Wager[51,52]. Green; the values fall in more desirable range, red: the values fall in less desirable range, yellow: the values fall in-between the ranges. ClogP: calculated logarithm of partition coefficient of a compound between n-octanol and water, ClogD: calculated logarithm of distribution coefficient of a compound between n-octanol and water at pH 7.4, MW: molecular weight, tPSA: topological polar surface area, CNS MPO: central nervous system multiparameter optimisation.

Along with ebselen derivatives, we also investigated the effect of ebsulphur and its analogues on the thermal stability of SOD1 (Fig. 2c, d, g, h). Despite ebsulphur-based compounds having similar chemical structures to ebselen-based compounds, the stability increases of both the WT$^{C6S}$ and A4V$^{C6S}$ are not attained, with only compounds **12**, **19** and **20**, which contain additional amide or ester linkers, showing a stabilising effect. We also used 4-acetamido-4'-maleimidylstilbene-2,2'-disulfonic acid (AMS) to observe modification of the free cysteines in A4V$^{C6S}$ SOD1 monitored by denaturing SDS-PAGE separation. The result shows that A4V$^{C6S}$-ebsulphur migrates to the same level as the control indicating Cys111 is not protected by reaction with ebsulphur. Conversely, A4V$^{C6S}$-ebselen shows a strong band equivalent to disulphide reduced and unmodified SOD1 thus indicating Cys111 is extensively protected by ebselen binding (Supplementary Fig. 2). These observations are consistent with the greater degree of reactivity of the ebselen core compared to the ebsulphur based core.

**Ebselen and its analogues covalently bind to A4V SOD1 at Cys111.** We recently described atomic-resolution crystal

structures of ebselen and ebsulphur bound to the Cys111 of wild-type SOD1 and showed that the compounds restored the mutant SOD1 post-translational modification pathway by promoting the formation of the intra-subunit disulphide bond and increasing dimer affinity[13]. Previously A4V has been crystallised in P2$_1$2$_1$2$_1$ and P2$_1$ space groups that have provided insight into the destabilisation of the A4V dimer[20,21] but in contrast to other mutants, no compound has ever been seen in A4V. In this study, we successfully co-crystallised A4V SOD1 on wild-type background (no C6S mutation) with various compounds in C2 space group and provide direct visualisation of the binding of ebselen, compounds **1**, **4** and **6** at Cys111. Crystallographic data collection and refinement statistics are presented in Table 2. The covalent bond distances between the selenium atom of each compound and sulphur atom of cysteine vary from 2.00–2.40 Å which can be due to partial bond breakage during x-ray data collection. The electron density of the ligand in many Cys111 binding sites is clearly visible except for some parts of the phenyl rings of ebselen, pyridine rings of compound **1**, chlorobenzyl ring of compound **4**, and the linker of compound **6** (Fig. 3). The absence of electron density is probably due to the flexibility of these parts of the ligands when exposed to solvent. Zn is in full occupancy in the Zn

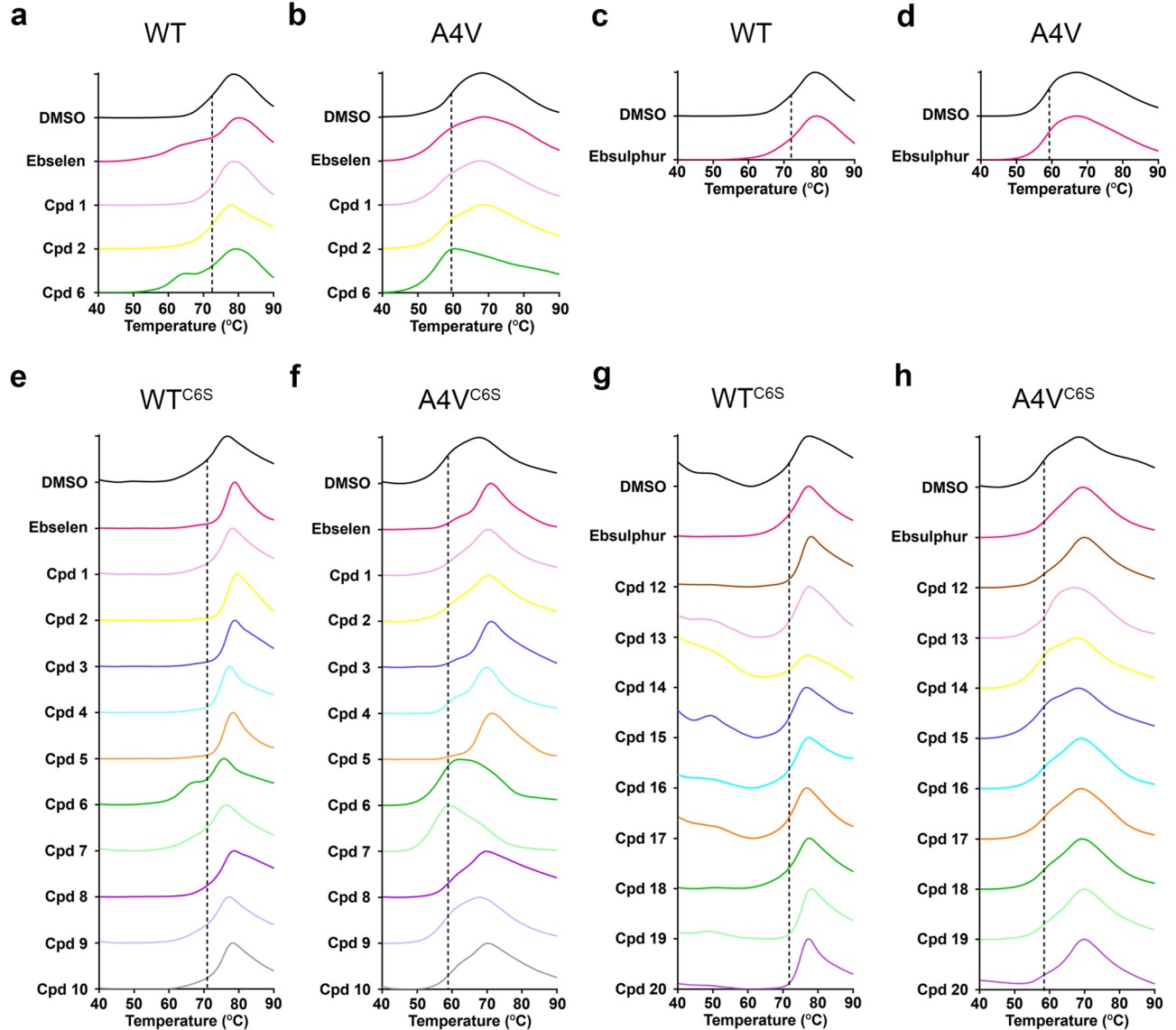

**Fig. 2 DSF screening of the small molecules bound to SOD1.** Normalised melting curves of the wild-type (WT) and A4V SOD1 with and without (**a**, **b**) ebselen derivatives and (**c**, **d**) ebsulphur. Normalised melting curves of WT$^{C6S}$ and A4V$^{C6S}$ with and without ebselen derivatives (**e**, **f**) and ebsulphur derivatives (**g**, **h**). Each graph is an average of 3 independent experiments. Black dashed lines show the T$_m$ (°C) of the control proteins: (**a** and **c**) WT SOD1; 72.4 °C, (**b** and **d**) A4V SOD1; 59.2 °C, (**e** and **g**) WT$^{C6S}$ SOD1; 70.7 °C, and (**f** and **h**) A4V$^{C6S}$ SOD1; 58.6 °C.

binding site (Supplementary Fig. 3a). No electron density for the compounds is seen around Cys6 (Supplementary Fig. 3b), the other free cysteine in SOD1, which might be due to slightly limited space and/or relatively weaker binding at Cys6. The disulphide bonds connecting Cys57 and Cys146 are intact (Supplementary Fig. 3c). This means electrophilic reaction of our selenium-containing compounds is preferable with Cys111 rather than other cysteines in crystalline state.

The binding poses of ebselen, compounds **1** and **6** observed in A4V SOD1 structures are similar in that the seleno-head group covalently binds at Cys111 and the aromatic tail extends into the surrounding solvent beyond the dimer groove (Fig. 3b, c, e). The structure of A4V-compound **4** shows a different binding pose, although the chemical structure is quite similar to ebselen with only an additional methylene linker. The seleno-head group of compound **4** is almost perpendicular to that of other compounds and the chlorobenzyl group extends along the dimer groove, filling the gap between N-termini and C-termini (Fig. 3d, f). Although the covalent bond is the main interaction connecting

the ligand and the protein, two ligands facing each other at the dimer interface are primarily linked by π-π stacking at aromatic rings (Fig. 4). This stacking interaction might play an important role in stabilising the dimer. Apart from the selenylsulphide bond, all structures show additional interactions between the ligands and the residues neighbouring Cys111, suggesting an asymmetrical binding of the compounds to the protein (Fig. 4). Water-bridging hydrogen bonds were observed in A4V-ebselen and A4V-compound **1** structures, indirectly connecting carbonyl group or aromatic ring of the ligands with Gly108 backbone or Asp109 side chain (Fig. 4a, b). The phenyl ring of ebselen also links with the peptide backbone of Gly108 and Asp109 through an amide-π interaction, which is also observed in our previous crystal structure of wild-type SOD1 conjugated with ebselen[13].

The A4V-compound **1** structure shows an additional hydrogen bond between the nitrogen atom of the pyridine ring and Gly108 backbone (Fig. 4b). The chlorobenzyl ring of compound **4** locates near Thr2 and interacts with Ile151 through a hydrophobic σ-π interaction, strengthening the binding of the compound by filling

the gap between N-termini and C-termini (Fig. 4c). In addition, the phenyl ring of compound **6** interacts with Asp109 side-chain oxygen atom through an anion-π interaction (Fig. 3d). Binding of

the ligands to A4V SOD1 does not seem to affect hydrogen bonding and hydrophobic interactions between two monomers of the dimer (Supplementary Table 1) or the distances between residues at the dimer interface except the distances between loop VI when comparing to the C2 A4V SOD1 structure (Supplementary Table 2).

**Monomer-monomer reorientation of ligand-conjugated A4V SOD1**. The A4V SOD1 substitution is known to change the monomer-monomer orientation within a SOD1 dimer[20]. To exclude the effect of differences imposed by crystal packing, the crystal structure of A4V SOD1 in C2 form was used to compare monomer-monomer orientation. The superposed structures of all C2 forms show high similarity with the range of root-mean-square deviation (RMSD) indicated in Fig. 5 and Table 3. The major difference is in loop VI, residues 109–114, which is more open in the compound-bound A4V structures (Fig. 5, box) with some variations among structures and an asymmetry of this loop between two monomers. The result suggests that loop VI is slightly displaced upon ligand binding.

The orientation of two monomers of ligand-conjugated A4V SOD1 was evaluated by superposition of monomer A of all dimers in the asymmetric unit of each structure with the wild-type SOD1 structure (Supplementary Fig. 4). The changes in RMSD, angle of rotation, and displacement of monomer B were then observed. Six dimers in the asymmetric unit (ASU) of the P2$_1$ A4V structure (PDB code 1UXM) were also compared with the wild-type SOD1 structure as a reference. The movement is complex, with a combination of rotation and displacement (Supplementary Fig. 4a). The comparison reveals that the changes in the orientation of two monomers of ligand-conjugated A4V SOD1 and C2 A4V structures are mostly within the range of those found in P2$_1$ A4V SOD1 crystal structures (Supplementary Fig. 4b), although the structures of A4V-compound **6** appear to

**Table 1 ΔT$_m$ of the proteins conjugated with ebselen-based and ebsulphur-based derivatives.**

| Compounds | ΔT$_m$ (°C) | | | |
|---|---|---|---|---|
| | **WT** | **A4V** | **WT$^{C6S}$** | **A4V$^{C6S}$** |
| Ebselen | −10.5* | −3.1 | 5.6 | 8.8 |
| Cpd **1** | 1.0 | −1.7 | 3.3 | 4.5 |
| Cpd **2** | −0.1 | −0.4 | 3.1 | 4.0 |
| Cpd **3** | – | – | 5.4 | 8.8 |
| Cpd **4** | – | – | 3.7 | 6.7 |
| Cpd **5** | – | – | 4.8 | 8.7 |
| Cpd **6** | −9.9* | −4.7 | −6.6* | −2.6 |
| Cpd **7** | – | – | −0.6 | −3.4 |
| Cpd **8** | – | – | 3.0 | 3.9 |
| Cpd **9** | – | – | 1.7 | 1.3 |
| Cpd **10** | – | – | 3.4 | 3.9 |
| Ebsulphur | −0.3 | −0.1 | 0.6 | 2.4 |
| Cpd **12** | – | – | 2.8 | 3.8 |
| Cpd **13** | – | – | 1.4 | 0.5 |
| Cpd **14** | – | – | 0.2 | −1.5 |
| Cpd **15** | – | – | 0.5 | −1.5 |
| Cpd **16** | – | – | 0.8 | 1.5 |
| Cpd **17** | – | – | 0.5 | 1.2 |
| Cpd **18** | – | – | 0.6 | 2.0 |
| Cpd **19** | – | – | 3.1 | 3.3 |
| Cpd **20** | – | – | 2.7 | 4.9 |

*ΔT$_m$ of the control and the first T$_m$ of the biphasic curve derived by calculating the peak of the first derivative of the raw data.
The figures are means of 3 independent experiments.

**Table 2 Crystallographic data collection and refinement statistics.**

| | A4V (C2) | A4V-ebselen | A4V-cpd 1 | A4V-cpd 4 | A4V-cpd 6 |
|---|---|---|---|---|---|
| Data collection | | | | | |
| Space group | C2 | C2 | C2 | C2 | C2 |
| Cell dimensions | | | | | |
| *a, b, c* (Å) | 112.60, 195.96, 75.68 | 112.31, 195.24, 75.35 | 112.51, 194.66, 75.17 | 113.53, 194.67, 146.52 | 112.66, 195.55, 76.34 |
| α, β, γ (°) | 90.00, 97.10, 90.00 | 90.00, 97.85, 90.00 | 90.00, 96.97, 90.00 | 90.00, 93.44, 90.00 | 90.00, 98.08, 90.00 |
| Resolution (Å)$^a$ | 97.98–1.65 (1.68–1.65) | 47.85–2.25 (2.3–2.25) | 97.33–1.97 (2.00–1.97) | 58.47–2.80 (2.86–2.80) | 97.77–2.77 (2.88–2.77) |
| $R_{merge}$$^a$ | 6.66 (100.46) | 8.4 (46.0) | 7.1 (36.6) | 17.4 (62.9) | 15.5 (59.3) |
| I/σI$^a$ | 11.4 (1.19) | 10.3 (1.9) | 6.7 (1.6) | 3.7 (1.3) | 4.7 (1.9) |
| CC$_{1/2}$ (%)$^a$ | 0.998 (0.465) | 0.996 (0.639) | 0.993 (0.776) | 0.965 (0.538) | 0.972 (0.674) |
| Completeness (%)$^a$ | 99.6 (99.5) | 97.3 (97.7) | 99.2 (95.5) | 99.7 (99.7) | 100.0 (100.0) |
| Redundancy$^a$ | 3.4 (3.5) | 3.5 (2.8) | 2.9 (2.0) | 3.1 (3.0) | 3.3 (3.4) |
| Refinement | | | | | |
| No. reflections | 183807 | 70706 | 105982 | 73673 | 138246 |
| $R_{work}$/$R_{free}$ | 16.89/19.61 | 18.59/22.26 | 18.28/22.61 | 21.32/24.11 | 21.18/22.89 |
| No. atoms | | | | | |
| Protein | 6,893 | 6,688 | 6,869 | 13,385 | 6,683 |
| Ligand/ion | 95/6 | 169/6 | 318/18 | 306/12 | 308/22 |
| Water | 1375 | 812 | 1015 | 230 | 240 |
| *B*-factors | | | | | |
| Protein | 23.24 | 18.59 | 32.15 | 31.82 | 48.22 |
| Ligand/ion | 37.36/17.77 | 28.62/15.20 | 46.0/29.38 | 85.56/23.72 | 61.99/31.21 |
| Water | 40.96 | 39.89 | 41.58 | 18.21 | 28.64 |
| R.M.S. deviations | | | | | |
| Bond lengths (Å) | 0.013 | 0.008 | 0.008 | 0.008 | 0.005 |
| Bond angles (°) | 1.780 | 1.476 | 1.451 | 1.353 | 1.295 |
| PDB code | 6SPA | 6SPH | 6SPJ | 6SPI | 6SPK |

$^a$Values in parenthesis denote the highest resolution shell. The data of each structure was collected from one single crystal.

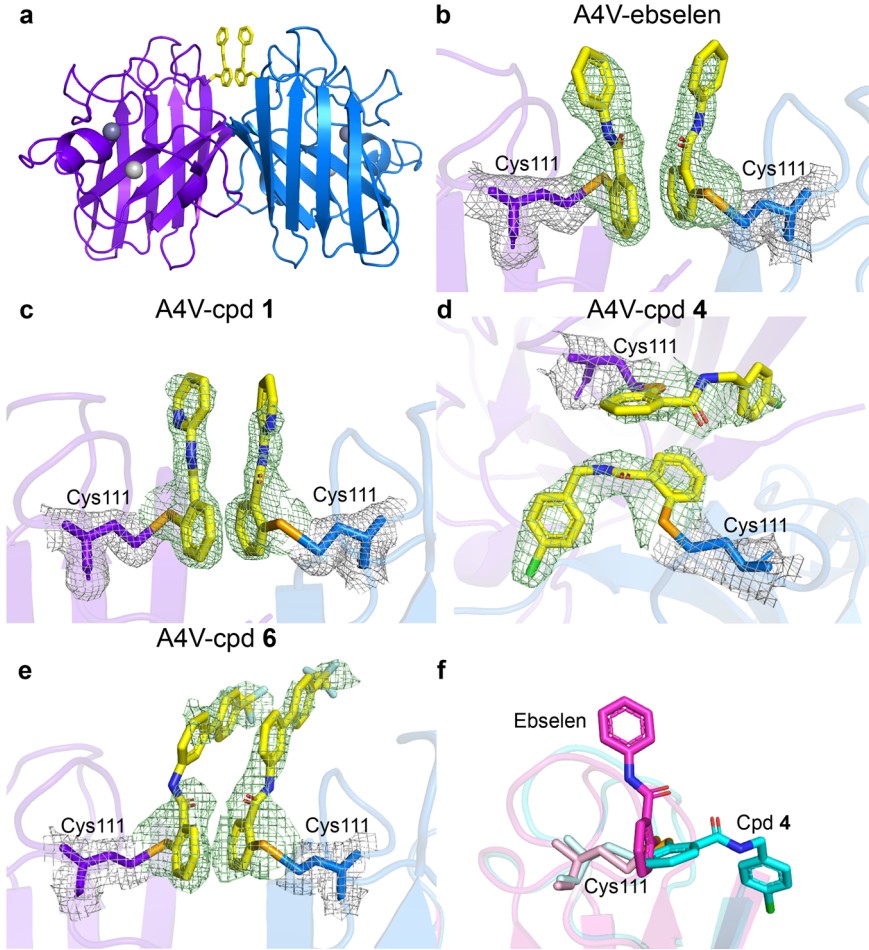

**Fig. 3 The structures of A4V SOD1 with bound ligands. a** Cartoon representation of A4V SOD1 dimer with ligands. The ligands (yellow sticks with valences) bind each monomer (purple and blue) at Cys111, located at the dimer interface. Dark grey spheres represent Zn in the Zn binding sites and light grey spheres represent Zn in the Cu binding site. Details of ligand binding of **b** A4V-ebselen, **c** A4V-compound **1**, **d** A4V-compound **4** and **e** A4V-compound **6**. The Fo-Fc map contoured at 3σ (green) shows the electron density of the ligands. The 2Fo-Fc map contoured at 1σ (grey) shows the electron density of Cys111 residues. **f** The superposed structures of A4V-ebselen (magenta) and A4V-compound **4** (cyan) monomers, depicting the binding pose of each compound.

shift away more than other ligand-bound A4V structures and some dimers of A4V-compound **4** structure moves towards the wild-type SOD1 structure, marginally further than A4V SOD1 without ligand. We note there is a structural variability across A4V SOD1 structures which is not eliminated by crystallisation and whether the subtle changes in the compound-bound A4V SOD1 structures are related to the dimer stability requires further investigation.

## Discussion

ALS is a result of complex aetiologies, although clinical manifestations of familial and sporadic types are similar. Both types of ALS share a common feature, which is cytosolic protein aggregations including SOD1[1]. The A4V substitution is one of the most severe SOD1 mutations and causes destabilisation and aggregation in fALS individuals. While the mechanism of dimer destabilisation and the increased propensity to aggregate remain elusive, several structural and molecular dynamics studies have demonstrated that the A4V mutation causes structural changes in neighbouring residues close to the dimer interface, which potentially perturb overall dimer stability even though the interface interactions are generally conserved in the crystal structure[20,22–24]. The network of contact

interactions between the dimer interface and Zn binding loop is also affected by A4V mutation. The disorder of the Zn-loop impairs the Zn binding ability of SOD1 which together with the instability of electrostatic loop, in turn, could alter the dimer interface[23–25]. As a consequence of dimer destabilisation, dissociation of the dimer was shown to cause exposure of the hydrophobic surface of each monomer, resulting in protein aggregation[7,26]. Therefore, stabilisation of dimeric A4V SOD1 could be a potent strategy for the development of ALS therapeutics.

Cys111 located at the edge of the dimer interface and exposed to the surrounding environment serves as a double-edged sword. The exposure of the thiol group of cysteine makes it prone to oxidative damage by physiological oxidants such as oxidised glutathione or thioredoxin[27,28]. This cysteine residue also plays a critical role in the formation of incorrect inter-subunit disulphide bonds, leading to toxic high molecular weight oligomers[29,30], since substitution of Cys111 with serine in SOD1 mutants, including A4V, and demetallated wild-type SOD1 could mitigate intracellular aggregation[29]. On the other hand, several studies have shown that Cys111 could be a potential target for the development of compounds to stabilise SOD1. Binding of small molecules to this residue does not only reduce the chance of oxidative modification, but also promotes dimer stability by tethering each monomer or providing additional

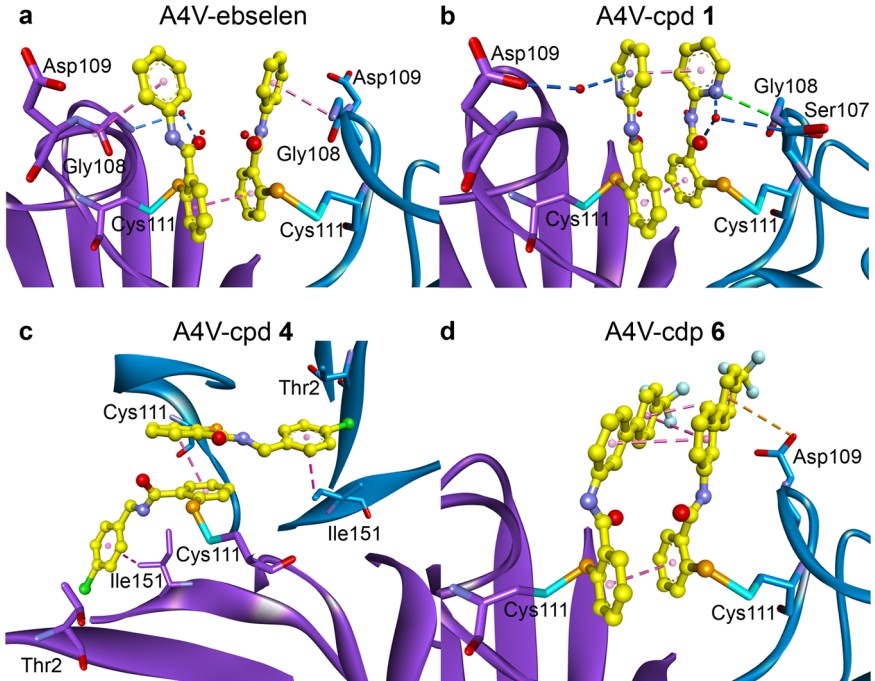

**Fig. 4 Ligand-ligand and ligand-A4V SOD1 interactions. a** A4V-ebselen, **b** A4V-compound **1**, **c** A4V-compound **4** and **d** A4V-compound **6**. The ligands are represented as yellow ball and stick, and waters are represented as red dots. The selenium atom (orange) of the ligand binds covalently to the sulphur atom (cyan) of Cys111. Hydrophobic interactions (pink dashed lines) are shown between the aromatic rings (π-π stacking) of the ligands bound at the dimer interface, between the phenyl ring of the ebselen and Gly108-Asp109 amide bond (amide-π stacking) of the A4V-ebselen structure, and between the chlorobenzyl group and Ile151 (σ-π interaction) of the A4V-compound **4** structure. An electrostatic anion-π interaction (orange dashed line) is shown between the aromatic ring of compound **6** and the side chain of Asp109. Water-bridged hydrogen bonds (blue dashed lines) are found in A4V-ebselen and A4V-compound **1** structures. An additional hydrogen bond (green dashed line) is also found between the nitrogen atom of the pyridine ring of compound **1** and Gly108 backbone amide.

interactions between two monomers[11–13]. Our study shows that when ebselen and its analogues bind only to Cys111 of A4V SOD1, they increase the thermal stability of the protein. The crystal structures of ligand-bound A4V mutant and wild-type SOD1[13] agree with the thermal shift study that the compounds form a covalent bond with the sulphur atom of Cys111 of each monomer and the aromatic rings are in the position which promotes face-to-face π–π stacking interaction. This hydrophobic interaction is ubiquitously present in biological macromolecules and important for protein-protein or protein-nucleic acid recognition. The aromatic ring of hydrophobic amino acids also contributes to protein folding and maintaining secondary structure, providing approximately 1 kcal/mol of energy to protein stability[31–34]. The extended dimeric contact by π–π stacking interaction in our crystallographic study might strengthen the existing hydrogen and hydrophobic interactions at the dimer interface, without changing interdimeric distance, and thereby increasing A4V SOD1 thermal stability.

Our thermal shift assay shows the differences in the melting temperature of the proteins upon binding to ligands. While the unfolding of proteins may not always indicate the aggregation process or the degree of dimer stability[9,10], this method is feasible for the screening of ligand-protein interactions[35]. The result of DSF showed that lower molecular mass compounds (ebselen and compounds **1–5**) could increase thermal stability of WT$^{C6S}$ and A4V$^{C6S}$ more than larger mass compounds (compounds **6–10**). The smaller compounds might fit in the dimeric groove better than the larger compounds without disturbing the interaction at the dimer interface and could form π–π stacking interaction as mentioned above. Notably, compounds **3**, **4** and **5**, which are methylene-linked analogues, greatly increased SOD1 melting temperatures. This result can be understood because the methylene linker makes the

aromatic tail group more flexible, allowing the chlorobenzyl ring to localise between N- and C-termini and interact with Ile151, which is a residue critical for maintaining hydrogen bonding between two monomers. Thus, the binding pose as demonstrated in A4V-compound **4** structure might serve as a bridge, reinforcing the interaction at the dimer interface.

Although increasing the number of aromatic rings of the small molecules is one of the methods to make the ligand more permeable to lipid bilayer of cell membranes, the increase in size and hydrophobicity could reduce its solubility and affect the binding affinity for the target protein[32,36]. The slight increases, or even decreases, in the melting temperature of compounds **6–10** in our study may be due to the steric restriction that prevents the head group from gaining access to the binding site, or the compounds in solution might adopt the binding pose that less favours π-π stacking interactions. The structure of compounds **6** and **7** contains ebselen as a core with the addition of phenyl ring and trifluoromethyl group resulting in relatively different properties from ebselen, as demonstrated in Fig. 1, especially their size and solubility. The decrease in T$_m$ of the protein when binding to compounds **6** and **7** may be because the biphenyl trifluoromethyl-containing compounds adopt a different binding pose or a different binding site from the crystal structure. The strong electronegativity of fluorine[37] may also interfere with the bonding at the dimer interface. In addition, the biphenyl tail group without methylene bridge in compounds **6** and **7** may increase hydrophobicity of the compounds with less flexibility than compounds **8**, **9** and **10**. The position and the binding site in solution, together with the strong electronegativity and rigidity, probably cause repulsion across the dimer interface, resulting in a shift of the monomer-dimer equilibrium toward monomer, which was

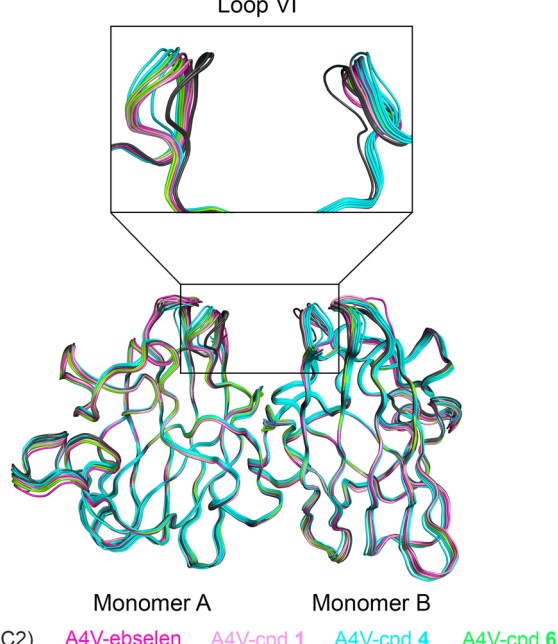

**Fig. 5 Superposition of A4V SOD1 C2 structures.** All dimers in the asymmetric unit (ASU) of each structure were overlaid onto one dimer of the A4V SOD1 structure without ligands. The average RMSD of three dimers in the ASU of the A4V SOD1 structure is 0.19 Å. The expanded box shows the detail of loop VI which is more opened in the compound-bound A4V SOD1 structures. The table shows the maximum and the minimum of RMSD of each ligand-conjugated structure compared to the structure without ligands.

**Table 3 The RMSD of the ligand-conjugated A4V SOD1 structures compared to the A4V SOD1 in C2 form structure without ligands.**

| Structures | No. of dimers in the ASU | RMSD (Å) |
|---|---|---|
| A4V-ebselen | 3 | 0.24–0.33 |
| A4V-cpd **1** | 3 | 0.21–0.32 |
| A4V-cpd **4** | 6 | 0.19–0.60 |
| A4V-cpd **6** | 3 | 0.30–0.40 |

The figures are ranges of the maximum and the minimum of the RMSD of each structure.

shown to have lower thermal stability[38]. However, our study shows that the compounds could not only bind at Cys111, but also Cys6 in solution which may be due to the hydrophobicity of the compounds that could penetrate into the β-barrel structure. We must remember that intracellular SOD1 is unlikely to encounter the high concentration of ebselen derivatives used for in vitro conjugation here, but this characteristic will be considered for the development of the next generation of ebselen analogues. To make the compounds more hydrophilic and less rigid, incorporating hydrophilic saturated heterocycles together with flexible linkers to ebselen would be interesting strategies for developing small compounds that react with Cys111 and form hydrogen bonds with the surrounding residues thereby increasing the ligand-ligand and ligand-protein interactions while maintaining the desirable CNS MPO score.

In spite of our best efforts, we could not obtain the crystal structure of A4V SOD1 with ebsulphur bound. The DSF result demonstrates that ebsulphur and its analogues also promote thermal stability of WT[C6S] and A4V[C6S], but to a much lesser extent than ebselen derivatives, and AMS modification of free cysteines

shows the mobility shift of the protein with ebsulphur which may be due to a stronger reactivity of selenium than sulphur[39]. This is consistent with our previous study, showing the smaller increase in dimer affinity when SOD1 mutants are bound to ebsulphur compared to ebselen[13]. Therefore, future development of SOD1-targeted compounds should focus mainly on selenium-containing or possibly tellurium-containing ebselen derivatives.

In conclusion, our study has developed the DSF assay using C6S background SOD1, which can provide medium-throughput evaluation of candidate lead compounds prior to co-crystallography and cellular evaluation. We present the promising capability of ebselen and its derivatives with molecular weight below 400 Da in promoting the thermal stability of A4V SOD1 when binding to Cys111 only. The binding site in A4V SOD1 of ebselen-based compounds has been validated by crystallographic study indicating all lead compounds form a covalent bond of Se–S to the side chain of Cys111. Small compact selenium compounds with flexible linkage, e.g., compounds **3**–**5**, are more potent candidates than sulphur-containing compounds in stabilising the SOD1 dimer. Although the crystal structures show the binding of compounds to Cys111 in the position which could facilitate π–π stacking interaction, it is possible that the large ligands have more rotational freedom in solution, which may disrupt the dimeric interaction and should be considered when designing new ligands. We note that none of the previous attempts for chemical intervention with A4V have yielded compounds in crystallographic structure of A4V[9,13]. This has hampered structure-based lead optimisation due to paucity of basic knowledge on the binding region and associated interactions critical for understanding binding space and setting a strategy for knowledge-based lead optimisation. However, to become a potential medication, our ebselen derivative series should be improved probably by combining interactions seen with different compounds while increasing solubility and specificity to eliminate the possibility of binding to Cys6. Therefore, this study provides a platform for using the ebselen analogue template for structure-based drug development for ALS therapeutics.

## Methods

**Synthesis of ebselen-based and ebsulphur-based compounds.** Details for the synthesis of ebselen and ebsulphur analogues are included in the supplementary information.

**Recombinant SOD1 production.** The mutant A4V SOD1 was generated by site-directed mutagenesis as previously described[13]. The mutant wild-type[C6S] (WT[C6S]) and A4V[C6S] SOD1s were also produced by site directed mutagenesis (Genscript). The pET303C plasmids containing wild-type, A4V, WT[C6S] and A4V[C6S] human *SOD1* genes were transformed into *E. coli* BL21 (DE3). Protein expression was induced with 0.4 mM Isopropyl β-d-1-thiogalactopyranoside (IPTG) supplemented with 0.3 mM ZnCl$_2$ and incubated at 18 °C overnight. Proteins were purified as previously described[40]. Briefly, proteins in 20 mM Tris-HCl pH 8.0 were purified using ion-exchange chromatography on diethylaminoethanol sepharose and eluted in a stepwise gradient concentration of NaCl. Proteins in the low NaCl concentration (5–100 mM) were pooled and subjected to size-exclusion chromatography on a Superdex 200 16/600 column in Tris-buffered saline (TBS) pH 7.4. The purified proteins were snap-frozen in liquid nitrogen and stored at −20 °C. The concentration of all SOD1 proteins was measured by ultraviolet spectrometry at 280 nm using a monomeric molar extinction of 5500 M$^{-1}$cm$^{-1}$.

**Crystallisation.** Ebselen and compound **6** were dissolved in DMSO for 150 mM stock solution which were then added to 3.9 mg/mL (250 μM) of A4V SOD1 to a final concentration of 1 mM. The mixtures were incubated at room temperature for 2 h before crystallisation. The crystals were grown using the hanging drop method by mixing 2 μL of the protein solution and 2 μL of reservoir solution containing 100 mM Tris-HCl pH 7.4–8.0, 2.4–2.6 M ammonium sulphate, 150 mM NaCl. The A4V SOD1 crystals in C2 space group were grown from the mixture of the protein and ebselen in the same conditions; however, no electron density of the compound was observed in the structure. Compound **1** and compound **4** were prepared from the same concentration of stock solution before adding to 5 mg/mL (300 μM) of A4V SOD1 to a final concentration of 500 μM. The mixtures were pre-incubated at 4 °C overnight before crystallisation. The crystals were grown using the hanging drop method by mixing 1 μL of the protein solution and 1 μL of reservoir solution

as mentioned above. All protein crystals were grown at 20 °C. The crystals were in mature size after 3–5 days and then soaked in 20% glycerol in reservoir solution before being snap-frozen in liquid nitrogen.

**Data collection and structural determination**. For A4V-ebselen crystal, data were collected at Barkla X-ray laboratory, University of Liverpool with 1.54 Å wavelength using Rigaku FR-E+ Super-Bright rotating anode generator with an EIGER R 4 M detector to 2.2 Å resolution at 100 K, followed by integration using *automar* software[41] and scaling using Aimless[42]. Other data were collected using the I04-1 beamline at the Diamond Light Source (Harwell, UK) at 100 K with x-rays of wavelength 0.92 Å, integrated using DIALS[43] or iMOSFLM[44] and scaled using Aimless[42]. All structures were solved by molecular replacement using MOLREP[45] software as a part of CCP4 program suite[46], and A4V SOD1 structure (1UXM; chain A and B) was used as a starting model. Restrained refinement in Refmac5[47,48] was iterated with model building in COOT[49]. In the final stages of refinement, TLS refinement was used to refine A4V-ebselen structure, and hydrogen atoms were added in riding position to all structures. The models of ebselen and ebselen-based compounds were produced in JLigand[50]. The structures were validated using MolProbity[50] before being deposited in the Protein Data Bank (PDB codes 6SPA, 6SPH, 6SPI, 6SPJ and 6SPK).

**Differential scanning fluorimetry (DSF)**. DSF was carried out using a StepOnePlus Real-Time PCR machine (Life Technologies). Proteins at 40 µM were pre-incubated with 100 µM of ebselen and ebselen analogues prepared from 150 mM stock solution at 4 °C overnight. The final concentration of DMSO in the mixture was 1.25% (v/v). Prior to DSF assay, the mixture was dialysed against TBS at 4 °C for 3 h to remove excess compounds. SYPRO Orange dye (Life Technologies) dye at 10× concentration was used as a probe. Fluorescent intensities were monitored from 25 °C to 95 °C at a ramp rate of 0.3 °C/min. The analysis of melting temperature ($T_m$) was based on the Boltzmann equation and calculated using the TmTool™ software (Life Technologies).

**Free-thiol assay**. A4V$^{C6S}$ SOD1 was pre-incubated with ebselen and ebsulphur as previously stated. After removal of excess compounds, the samples were blocked with 100-fold excess of 4-acetamido-4′-maleimidylstilbene-2,2′-disulfonic acid (AMS) for 2 h at 37 °C before being separated by denaturing 15% SDS-PAGE.

**Statistics and reproducibility**. In DSF assay, data were taken from 3 independent experiments with 4–6 replicates per experiment. Each protein was from a single batch of purification. In crystallographic study, high multiplicity could be obtained from data collection of a single crystal.

**Reporting summary**. Further information on research design is available in the Nature Research Reporting Summary linked to this article.

## Data availability

Crystal structures of A4V SOD1 with compounds are deposited in the Protein Data Bank under accession number 6SPA, 6SPH, 6SPI, 6SPJ, and 6SPK. Other relevant data are available upon request from the corresponding author.

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

## Acknowledgements
This work is supported by funding from ALS Association (USA) to P.M.O.N and S.S.H. (WA-1128). We would like to thank the staff and management of the Diamond Light Source, especially staff of the beamline I04-1, for their help and smooth operation of the facility. We thank members of the Liverpool's Molecular Biophysics Group for the help in data collection and their interest in the project. We would also like to thank Dr. Patrick Ridley, University of Liverpool for his interest. M.S. was supported by the Higher Education Commission Pakistan for six-month research training visit to Liverpool. V.C. is supported by Mahidol-Liverpool PhD Scholarship.

## Author contributions
S.S.H., S.V.A., P.M.O.N. and G.S.A.W. designed research; V.C., K.A., M.S., G.W., M.R., N.R. and M.P. performed research; M.S., G.W., M.R., N.R., M.P. and P.M.O. contributed new reagents; V.C., G.S.A.W., K.A., M.S. and S.V.A. analysed data; and V.C., G.S.A.W., K.A., S.V.A., P.M.O. and S.S.H. wrote the paper.

## Competing interests
The authors declare no competing interests.
