## [Peer Review File · Communications Biology]

Reviewers' comments:

Reviewer #1 (Remarks to the Author):

SOD1 is stable as a dimer, but mutations to the protein (such as A4V) increase aggregation that can only occur if the dimer dissociates into monomers. The main interest of this paper, to discover a small molecule which stabilizes the dimeric form of SOD1 mutants, is an in depth expansion on previous failed attempts to discover a similar small molecule. The paper and figures were intelligible, and making the compound synthesis methods supplementary allows those not in the field to grasp the content without getting discouraged. The study implemented differential scanning fluorescence and X-ray crystallization analysis to determine the change in thermostability of dimeric SOD1 due to the addition of different ebselen based compounds. The author's main finding was that ebselen and similar compounds (which were synthesized for the study) increased the thermostability of both WT and A4V mutant SOD1 as long as the second free cysteine (Cys6) was removed using a point mutation (C6S). Due to the condition that the other free cysteine must be removed for ebselen to make any difference, the author concludes that ebselen can be used as a template for structure based drug development of ALS therapeutics.

1. Why is the CNS MPO desirability score range 2.6 to 6.0? Is this a standard for this algorithm or was there a reason this range was selected?

2. The title of does not summarize the paper. A4V mutant dimer of superoxide dismutase is already known to be involved in motor neuron disease due to its increased propensity to aggregate, so stating that the stabilization of the mutant dimer is also involved is not presenting a new idea. The study centralizes around ebselen's effect on the mutant dimer, so that should be conveyed in the title.

3. The introduction mentions that maleimide-derived compounds and cisplatin were used in previous studies, but they both prevented the complex formation between SOD1 and its copper chaperone (hCCS). The author explains ebselen's neuroprotective effects and seems to imply that since ebselen has shown these positive effects it could not be affecting the SOD1-hCCS complex. But, the author does not elaborate or cite anything which shows ebselen is a better option to focus on compared to maleimide-derived compounds or cisplatin. Is there any proof that ebselen is the best choice to pursue and does not have the same problems of previous potential compounds?

4. The entire paper skims over the necessity of the C6S mutation for ebselen and similar compounds to increase thermostability. Both the results and discussion sections avoid this by calling ebselen a template for structure based drug development without elaborating on what aspects of the compound would need to be different for it to potentially work as an ALS therapeutic. This should be expanded on in the discussion section as the next step of this study. Supplementary tables 1 and 2 could be discussed further as other possible residues to target using compounds similar to ebselen and the same methodology used in this study. Along with this, Table 2 and Figure 1 should include the change in T_m values and DSF screening for the proteins without the C6S mutation, or it should be clearer when discussing the table and figure that the same results were not seen without the mutation.

5. Figure 4 within the results section has an expanded box showing the variations at the electrostatic loop. Is this electrostatic loop the same as loop VI residues 109-114 or the Zinc binding loop (mentioned in the discussion)? If not then what is the significance of highlighting that loop? It is important to evaluate the structural differences the bound ligand is causing in comparison to the WT to see if the ligand is somehow causing the mutant to be closer in structure to the WT, but the importance of this experiment is not clearly explained and has to be inferred through the results. Besides the end conclusion from the experiment, that the ligand is not causing the mutant structure to

be closer to the WT structure, the figure seems to be less important to the study than supplementary figure 2 and it might be better to switch the two figures.

6. The author emphasizes multiple times that this study is the first time anyone has been able to crystallize a compound within A4V, but does not say what is different that made their study successful when others were not. Is ebselen the reason that they can crystallize the protein or is it a combination of ebselen and the mutation?

Reviewer #2 (Remarks to the Author):

This is an very compelling paper. The major strength is that this is a comprehensive and original "structural pharmacology" study based around the literature-validated hypothesis of inhibiting monomerization of SOD1 to treat SOD1-familial ALS, particularly the malignant mutation A4V. The few weaknesses for this reviewer include a lack of discussion on potential "off-pathway" reactivity of the ebselen-related compounds, including an anti-therapeutic activity to "hinder maturation of SOD1 by preventing complex formation between SOD1 and hCCS" as the authors note occurred with cisplatin. It is possible that the authors have generated this information in previous ebselen publications, which should be cited in the present submission. A very minor suggestion is that the word "deterioration" in the introduction (referring to motor neurons in ALS) should be replaced with "degeneration" -- deterioration does not adequately capture motor neuron cell death.

Reviewer #3 (Remarks to the Author):

In this manuscript, the authors have examined effects of ebselen and its derivatives on the structure and thermostability of A4V-mutant SOD1 proteins, which are causative for familial ALS. The authors already published the essence of their research on SOD1 and ebselen (ref. 13), and the current manuscript does not add new information that deserves to be published in this journal. Therefore, I do not recommend publication of this manuscript in *Communications Biology*; instead, this manuscript is better published in more specialized journal. Several major points are as follows;

There is no information on the metal-binding status and also the thiol-disulfide status in recombinant SOD1 proteins used for DSF and crystallization in this study. The authors should quantitate the copper and zinc contents in the SOD1 (WT and A4V) proteins and also confirm the presence of the conserved disulfide bond.

Efficiency of the modification with ebselen should be compared with that of ebsulphur. The authors have assumed that the ebsulphur cannot stabilize SOD1 because of its lower reactivity than that of ebselen; but such difference in the reactivities can be experimentally evaluated (e.g. mass spec).

The reaction of SOD1 (both WT and A4V) with ebselen was found to produce a heterogenous population of the thermally destabilized species, and the authors found that such a destabilized species were not detected when ebselen (and its derivatives) reacted with C6S-mutant SOD1 proteins. Does this mean that ebselen can react with Cys6 and also that SOD1 modified at Cys6 with ebselen is destabilized? Reactions of C111S as well as C6S/C111S with ebselen and its derivatives should be examined.

The authors have revealed crystal structures of SOD1 (A4V) modified with ebselen and its derivatives.

Does the A4V protein used for crystallization has Cys at position 6? If so, interpretation of the results are inconsistent with the experimental results that no stabilization (or even destabilization) of the A4V protein was observed by addition of ebselen (Fig. 1A). If the authors have used the A4V/C6S proteins for crystallization, they should clearly indicate it in their manuscript.

According to the crystal structural analysis, ebselen and three compounds appear to interact with (modify) the A4V (or A4V/C6S?) proteins in a quite similar way. Nonetheless, their effects on the thermostability of the A4V were sharply different, while the authors put a sentence on this difference in the Discussion section. It is quite important to understand why compound 6 destabilizes the protein albeit in a quite similar structure to that of ebselen. For development of the ALS drugs, such information is exclusively important.

Reviewer 1

Comment 1. Why is the CNS MPO desirability score range 2.6 to 6.0? Is this a standard for this algorithm or was there a reason this range was selected?

Response: The CNS MPO score range between 2.6 to 6 for our compounds in the paper. The MPO score equally accounts for all physicochemical properties calculated in Table 1. Each physicochemical property has a desirable range and if not met then the score is lowered and therefore effects the overall score when all physicochemical properties are accounted in the MPO score. For compounds to have a good MPO score, a value of 4.0 or higher is desirable for the compounds to cross the blood-brain barrier and reach our target. We have modified the text with reference to Table 1 in the revised manuscript.

Comment 2. The title of does not summarize the paper. A4V mutant dimer of superoxide dismutase is already known to be involved in motor neuron disease due to its increased propensity to aggregate, so stating that the stabilization of the mutant dimer is also involved is not presenting a new idea. The study centralizes around ebselen's effect on the mutant dimer, so that should be conveyed in the title.

Response: We have modified the title to "Stabilization of A4V mutant dimer: Development of ebselen as template for motor neuron disease therapy using co-crystallography".

Comment 3. The introduction mentions that maleimide-derived compounds and cisplatin were used in previous studies, but they both prevented the complex formation between SOD1 and its copper chaperone (hCCS). The author explains ebselen's neuroprotective effects and seems to imply that since ebselen has shown these positive effects it could not be affecting the SOD1-hCCS complex. But, the author does not elaborate or cite anything which shows ebselen is a better option to focus on compared to maleimide-derived compounds or cisplatin. Is there any proof that ebselen is the best choice to pursue and does not have the same problems of previous potential compounds?

Response: We have demonstrated in a previous publication that ebselen does not prevent complex formation between SOD1 and hCCS¹. The advantage of ebselen over maleimide-derived compounds and cisplatin has been mentioned in the revised manuscript. We now have included these observations as part of the Introduction.

Comment 4. The entire paper skims over the necessity of the C6S mutation for ebselen and similar compounds to increase thermostability. Both the results and discussion sections avoid this by calling ebselen a template for structure-based drug development without elaborating on what aspects of the compound would need to be different for it to potentially work as an ALS therapeutic. This should be expanded on in the discussion section as the next step of this study. Supplementary tables 1 and 2 could be discussed further as other possible residues to target using compounds similar to ebselen and the same methodology used in this study. Along with this, Table 2 and Figure 1 should include the change in T_m values and DSF screening for the proteins without the C6S mutation, or it should be clearer when discussing the table and figure that the same results were not seen without the mutation.

Response: We have elaborated on the necessity of C6S mutation in the Discussion section of the revised manuscript and included our thoughts on compound development to overcome this off-target binding. The changes in T_m of the protein without C6S mutation in the presence of ebselen and some of its derivatives are in Table 2. We have also amended Figure 1 to show the change in melting curves. We have added the sentence regarding these changes in the revised manuscript as the reviewer suggested.

Comment 5. Figure 4 within the results section has an expanded box showing the variations at the electrostatic loop. Is this electrostatic loop the same as loop VI residues 109-114 or the Zinc binding loop (mentioned in the discussion)? If not then what is the significance of highlighting that loop? It is important to evaluate the structural differences the bound ligand is causing in comparison to the WT to see if the ligand is somehow causing the mutant to be closer in structure to the WT, but the importance of this experiment is not clearly explained and has to be inferred through the results. Besides the end conclusion from the experiment, that the ligand is not causing the mutant structure to be closer to the WT structure, the figure seems to be less important to the study than supplementary figure 2 and it might be better to switch the two figures.

Response: The electrostatic loop (loop VII; residues 122-144), the Zn binding loop (loop IV; residues 49-83), and loop VI (residues 102-115) are different. Our original Figure 4 focused on the electrostatic loop because this loop is the furthest region in monomer B and allows us to observe the difference among this monomer when monomer A is superposed although this difference could also be observed in other loop regions of monomer B. However, there are variations among each dimer in the asymmetric unit and the range of variation overlaps

among A4V SOD1 crystal structures. The ordering of both electrostatic and zinc loops are critical for stability of the dimeric SOD1.

Figure 4 and former supplementary figure 2 have been switched and this section has been re-written in the revised manuscript as the reviewer suggested.

Comment 6. The author emphasizes multiple times that this study is the first time anyone has been able to crystallize a compound within A4V but does not say what is different that made their study successful when others were not. Is ebselen the reason that they can crystallize the protein or is it a combination of ebselen and the mutation?

Response: The behaviour and properties of WT and A4V SOD1 differ. Inherent structural instability of A4V SOD1 has traditionally proven difficult to crystallise. Since our initial success in 2004 that defined the A4V structure for the first time, the particular crystal form did not yield diffracting when co-crystallographic experiments have been performed with compounds. This was one reason we were not able to show the effect of ebselen on A4V in last year's NCOMMS publication (reference 1). The new crystal form yields well diffracting crystals of A4V (no C6S mutation) in co-crystallographic experiments. We want to emphasise that it is not just others who had not succeeded with A4V co-crystallographic experiment we had not succeeded either until now.

Reviewer 2

Comment: This is a very compelling paper. The major strength is that this is a comprehensive and original "structural pharmacology" study based around the literature-validated hypothesis of inhibiting monomerization of SOD1 to treat SOD1-familial ALS, particularly the malignant mutation A4V. The few weaknesses for this reviewer include a lack of discussion on potential "off-pathway" reactivity of the ebselen-related compounds, including an anti-therapeutic activity to "hinder maturation of SOD1 by preventing complex formation between SOD1 and hCCS" as the authors note occurred with cisplatin. It is possible that the authors have generated this information in previous ebselen publications, which should be cited in the present submission. A very minor suggestion is that the word "deterioration" in the introduction (referring to motor neurons in ALS) should be replaced with "degeneration" -- deterioration does not adequately capture motor neuron cell death.

Response: Please see point 3 above. The advantage of ebselen over maleimide-derived compounds and cisplatin has been mentioned in the revised Introduction. The word “deterioration” has also been changed to “degeneration”.

Reviewer 3

Comment 1. There is no information on the metal-binding status and also the thiol-disulfide status in recombinant SOD1 proteins used for DSF and crystallization in this study. The authors should quantitate the copper and zinc contents in the SOD1 (WT and A4V) proteins and also confirm the presence of the conserved disulfide bond.

Response: The protein was expressed in LB media supplemented with 0.3 mM ZnSO₄ but copper was not supplemented during expression or incorporated during purification. Our crystal structure density maps show very clear and intact disulphide bond of Cys57/146 and full occupancy zinc (Supplementary fig. 3). These data will be available from the Protein Data Bank depositions upon publication.

Comment 2. Efficiency of the modification with ebselen should be compared with that of ebsulphur. The authors have assumed that the ebsulphur cannot stabilize SOD1 because of its lower reactivity than that of ebselen; but such difference in the reactivities can be experimentally evaluated (e.g. mass spec).

Response: We have demonstrated that ebselen is more reactive to SOD1 Cys111 than ebsulphur using AMS modification of free thiols leading to ~ 0.5 kDa band shift in non-reducing SDS-PAGE. The result shows that A4V^{C6S}-ebsulphur migrates to the same level as the control indicating Cys111 is not protected by reaction with ebsulphur. Conversely, A4V^{C6S}-ebselen shows a strong band equivalent to disulphide reduced and unmodified SOD1 thus indicating Cys111 is extensively protected by ebselen binding. These results have been added to the supplementary information in the revised manuscript as Figure S2.

Comment 3. The reaction of SOD1 (both WT and A4V) with ebselen was found to produce a heterogenous population of the thermally destabilized species, and the authors found that such a destabilized species were not detected when ebselen (and its derivatives) reacted with C6S-mutant SOD1 proteins. Does this mean that ebselen can react with Cys6 and also that SOD1 modified at Cys6 with ebselen is destabilized? Reactions of C111S as well as C6S/C111S with ebselen and its derivatives should be examined.

Response: We have performed additional DSF assay on C111S SOD1 with ebselen and ebsulphur. The result of C111S SOD1 with ebselen shows the biphasic melting curve with

the first T_m decreasing from the control by ~ 10 °C. In contrast, ebsulphur does not change the T_m of the protein. These data have been added to the supplementary information in the revised manuscript as Figure S1.

Comment 4. The authors have revealed crystal structures of SOD1 (A4V) modified with ebselen and its derivatives. Does the A4V protein used for crystallization has Cys at position 6? If so, interpretation of the results are inconsistent with the experimental results that no stabilization (or even destabilization) of the A4V protein was observed by addition of ebselen (Fig. 1A). If the authors have used the A4V/C6S proteins for crystallization, they should clearly indicate it in their manuscript.

Response: We crystallised A4V SOD1 without C6S mutation. All crystal structures show that free Cys6 is tightly packed inside the β -barrel and no compound density is observed (Supplementary Figure 3). Conversely, the native protein in solution may not be as tightly packed as in the crystal and possibly allow the compounds to gain an access to Cys6.

The C6S mutant was thus created to study the effect of the compounds bound only to Cys111, as observed in our crystal structures. Our data provide clear guidance that ebselen and its analogues promote thermostability of A4V SOD1 when binding to Cys111 only. The DSF data on C6S background provides a convenient assay giving the possibility of using it to obtain a ranking score for the test compounds. However, we have elaborated on the necessity of C6S mutation in the Discussion section of the revised manuscript and included our thoughts on compound development to overcome this off-target binding.

Comment 5. According to the crystal structural analysis, ebselen and three compounds appear to interact with (modify) the A4V (or A4V/C6S?) proteins in a quite similar way. Nonetheless, their effects on the thermostability of the A4V were sharply different, while the authors put a sentence on this difference in the Discussion section. It is quite important to understand why compound 6 destabilizes the protein albeit in a quite similar structure to that of ebselen. For development of the ALS drugs, such information is exclusively important.

Response: Although the structure of compound 6 contains ebselen as a core with the addition of phenyl ring and trifluoromethyl group, its properties are relatively different from ebselen, as demonstrated in Table 1, especially its size and solubility. The decrease in T_m of the protein when binding to compound 6 may be because the biphenyl trifluoromethyl-containing compounds adopt a different binding pose. The strong electronegativity of fluorine² may also

interfere with the bonding at the dimer interface. In addition, biphenyl tail group without methylene bridge in compound **6** may increase hydrophobicity of the compounds with less flexibility than compounds **8**, **9**, and **10**. The extended length of these compound may make them more flexible in solution than ebselen when bound to SOD1 interface. The strong electronegativity may also contribute to repulsion across the dimer interface, resulting in a shift of the monomer-dimer equilibrium toward monomer, which was shown to have lower thermal stability³. We have included this in the Discussion section of the revised manuscript.

References

- 1 Capper, M. J. *et al.* The cysteine-reactive small molecule ebselen facilitates effective SOD1 maturation. *Nat Commun* **9**, 1693, doi:10.1038/s41467-018-04114-x (2018).
- 2 Patrick, G. L. *An introduction to medicinal chemistry*. 6th ed. edn, (Oxford University Press).
- 3 Vassall, K. A. *et al.* Decreased stability and increased formation of soluble aggregates by immature superoxide dismutase do not account for disease severity in ALS. *Proceedings of the National Academy of Sciences of the United States of America* **108**, 2210-2215, doi:10.1073/pnas.0913021108 (2011).

REVIEWERS' COMMENTS:

Reviewer #1 (Remarks to the Author):

The author fully addressed all comments and edited the manuscript to clarify any confusing areas. The revised title and revisions to the discussion section greatly clarify the importance of this study as a template for future research.

Reviewer #3 (Remarks to the Author):

In the revised manuscript, the authors added some information on the higher reactivity of ebselen to SOD1 than that of ebsulphur and also provided experimental evidence supporting that ebselen could destabilize SOD1 by reacting with Cys6. Ebselen and the other compounds can increase the thermal stability of SOD1 when bound at Cys111; however, those compounds could destabilize the protein when bound at Cys6. This means that those compounds are not appropriate for "template for motor neuron disease therapy". The authors have described their idea on how to reduce the reactivity of those compounds to Cys6; therefore, they should prepare such compounds and test their efficacy and selectivity to Cys111.

As pointed out in my previous comments, the authors already published the essence of their research on SOD1 and ebselen, and the current manuscript does not add new information that deserves to be published in this journal. Therefore, I do not recommend publication of this manuscript in *Communications Biology*; instead, this manuscript is better published in more specialized journal.